# A Pathogenic Role of Non-Parenchymal Liver Cells in Alcohol-Associated Liver Disease of Infectious and Non-Infectious Origin

**DOI:** 10.3390/biology12020255

**Published:** 2023-02-06

**Authors:** Kusum K. Kharbanda, Shilpa Chokshi, Irina Tikhanovich, Steven A. Weinman, Moses New-Aaron, Murali Ganesan, Natalia A. Osna

**Affiliations:** 1Research Service, Veterans Affairs Nebraska-Western Iowa Health Care System, Omaha, NE 68105, USA; 2Department of Internal Medicine, University of Nebraska Medical Center, Omaha, NE 68198, USA; 3Department of Biochemistry & Molecular Biology, University of Nebraska Medical Center, Omaha, NE 68198, USA; 4Institute of Hepatology, Foundation for Liver Research, London SE5 9NT, UK; 5Faculty of Life Sciences and Medicine, King’s College London, London SE5 8AF, UK; 6Department of Internal Medicine, University of Kansas Medical Center, Kansas City, MO 66160, USA; 7Research Service, Kansas City Veterans Administration Medical Center, Kansas City, MO 64128, USA

**Keywords:** alcohol-associated liver disease, liver macrophages, hepatic stellate cells, hepatocytes, T-cells, HIV

## Abstract

**Simple Summary:**

Over 70% of the liver comprises of parenchymal cells (named hepatocytes) and the rest 30% of cells are the non-parenchymal cells, which include macrophages (resident Kupffer cells), hepatic stellate cells, endothelial and immune cells. Alcohol consumption can cause liver injury, which is known as alcohol-associated liver disease (ALD). ALD development is partially based on the activation of non-parenchymal liver cells. In this review, we will address the mechanisms of ALD progression and will analyze the contribution of most of the liver non-parenchymal cells to alcohol-induced liver damage.

**Abstract:**

Now, much is known regarding the impact of chronic and heavy alcohol consumption on the disruption of physiological liver functions and the induction of structural distortions in the hepatic tissues in alcohol-associated liver disease (ALD). This review deliberates the effects of alcohol on the activity and properties of liver non-parenchymal cells (NPCs), which are either residential or infiltrated into the liver from the general circulation. NPCs play a pivotal role in the regulation of organ inflammation and fibrosis, both in the context of hepatotropic infections and in non-infectious settings. Here, we overview how NPC functions in ALD are regulated by second hits, such as gender and the exposure to bacterial or viral infections. As an example of the virus-mediated trigger of liver injury, we focused on HIV infections potentiated by alcohol exposure, since this combination was only limitedly studied in relation to the role of hepatic stellate cells (HSCs) in the development of liver fibrosis. The review specifically focusses on liver macrophages, HSC, and T-lymphocytes and their regulation of ALD pathogenesis and outcomes. It also illustrates the activation of NPCs by the engulfment of apoptotic bodies, a frequent event observed when hepatocytes are exposed to ethanol metabolites and infections. As an example of such a double-hit-induced apoptotic hepatocyte death, we deliberate on the hepatotoxic accumulation of HIV proteins, which in combination with ethanol metabolites, causes intensive hepatic cell death and pro-fibrotic activation of HSCs engulfing these HIV- and malondialdehyde-expressing apoptotic hepatocytes.

## 1. Introduction

Excessive alcohol consumption is a global healthcare problem with enormous social, economic, and clinical consequences. While chronic and heavy alcohol consumption causes structural damage and/or disrupts normal organ function in virtually every tissue of the body, the liver sustains the greatest damage, since it is the primary site of ethanol metabolism. Chronic and heavy alcohol consumption disrupts normal liver function and eventually causes hepatic structural damage, resulting in alcohol-associated liver disease (ALD).

In the liver, the most potent ethanol-metabolizing cells are hepatocytes. These parenchymal cells account for ~70–80% of the liver mass [1] and are the first cell type to sustain an alcohol-induced injury for the initiation of the disease process. However, it is the other ~20–30% of the liver mass, lumped together as the non-parenchymal cells (NPCs), which promote the progression of liver disease. These NPCs are the Kupffer cells, sinusoidal endothelial cells, hepatic stellate cells, and immune cells. This review deliberates the effects of alcohol on the activity and properties of NPCs, which are either residential or infiltrated into the liver from the general circulation. These cells play a pivotal role in the regulation of organ inflammation and fibrosis, both in the context of hepatotropic infections and in non-infectious settings. We will also address the important alcohol-induced alterations in the functions of some liver NPCs, such as macrophages, T cells, and hepatic stellate cells, which contribute to alcohol-associated liver disease progression. In addition to alcohol, the behavior of these cells is regulated by other secondary hits, such as gender and infection, etc.

## 2. Role of Liver Macrophages in ALD

There is a longstanding recognition of the critical role that macrophages play in the inflammatory response to alcohol. The first suggestion of this role was from early studies of Thurman and colleagues, who used techniques that either ablated the total liver macrophages [2] or blocked macrophage inflammatory signaling using TLR4 KO mice [3]. These studies showed that removing inflammatory macrophage function from the liver greatly reduced alcohol-induced liver inflammation and overall pathology. This led to the idea that although Kupffer cells are non-inflammatory under basal conditions, they become “activated” by alcohol and subsequently drive liver injury [2,4,5]. Subsequent studies showing that alcohol induces bacterial translocation from the intestine [6,7] further supported this idea that alcohol initiates a gut-to-liver crosstalk causing a pro-inflammatory response of the hepatic macrophages that produce alcohol-associated liver disease [8,9,10].

Over the years, our understanding of the nature and diversity of liver macrophages has dramatically increased. In particular, new developments in single-cell technologies have expanded our understanding of the situation in normal livers and in disease processes. More recently, studies have begun to provide insight into the diversity and evolution of liver macrophage changes that specifically occur in ALD. Nonetheless, detailed functional information about specific macrophage populations has been difficult to obtain, and we currently have a situation where the knowledge of transcriptome-defined macrophage subsets has outstripped our understanding of the functions of these subsets. Nonetheless, new information about their function has recently emerged, and thus, a new picture of liver macrophages in ALD is beginning to emerge.

Alcohol consumption induces changes in liver macrophage populations that appear to play a key role in inflammatory liver injury, fibrosis, and disease resolution [11,12,13]. One of the problems in studying these changes is the multiplicity of mouse models, many of which do not closely resemble the pathology in humans with ALD. In one commonly used diet, the Lieber–DeCarli liquid diet for alcohol administration [14], there are several phases of macrophage changes. Within the first 3 days of alcohol exposure, there is a burst of KC apoptosis followed by the rapid restoration of the KC population within a week [15]. Work from the Weinman lab demonstrated that this initial apoptosis burst was induced by alcohol dependent S574 phosphorylation of the transcription factor FOXO3, changing its transcriptional specificity to promote apoptosis [16,17]. Kupffer cell loss was associated with the entry of infiltrating MΦs and a transient increase in the LPS sensitivity. After 10 days of continued ethanol exposure, adaptations seemed to occur. The LPS sensitivity was reduced back to baseline, the Kupffer cell numbers were restored, and intrahepatic MΦs mRNA expression showed the prominence of anti-inflammatory cytokines and anti-inflammatory MΦ markers. These data suggest that MΦ populations in the liver evolve dynamically after alcohol exposure.

Longer-term exposure to the LD diet shows that the IM populations evolve as well. After 4 weeks of Lieber–DeCarli alcohol feeding in mice, the total KC numbers are decreased, and circulating Ly6C+ monocytes enter the liver where they differentiate into MΦs with at least two different phenotypes, possessing pro- or anti-inflammatory gene expression patterns [11,18,19]. Subsequent studies showed that the KCs were not the dominant inflammatory cells and that a subfraction of the IMs, the recently recruited Ly6C high IMs, were largely responsible for inflammation [19]. In humans, immunohistochemistry [20] and gene deconvolution approaches [21] have shown the diversity of macrophage phenotypes in advanced alcohol-associated liver disease with the expression of multiple different subsets of both inflammatory and non-inflammatory macrophage populations.

To better identify the functions of macrophage populations in alcohol, we used a new mouse model for ALD that more closely resembles ASH [22]. This involves 16-week exposure of the mice to a high-fat chow diet supplemented with 10–20% alcohol in the drinking water. This treatment generates severe steatosis, inflammation, ballooning degeneration, and zone 3 pericellular fibrosis, which is histologically similar to moderately severe alcohol-associated steatohepatitis [22]. A single-cell RNA sequence analysis of liver macrophage populations from this model identified multiple KC and IM populations that differed from those in chow-fed mice. The bulk of the alcohol-associated KCs were non-inflammatory and expressed genes related to scavenger receptors, endocytosis, and lipid metabolism. Less abundant KC subsets were present as well and these had transcriptomic signatures that predicted functions associated with inflammation and extracellular matrix degradation. While the total KC numbers were similar in the chow- and alcohol-fed mice, IMs were much more abundant in alcohol-fed mice, where their abundance increased from about 10% to more than 50% of the total liver macrophages. IMs were also heterogeneous with one of the largest subsets, showing a classic pro-inflammatory signature characterized by NF-κB activation and pro-inflammatory cytokine production.

It has been difficult to establish the function of liver macrophage subsets by in vitro assessments. One of the problems is that macrophage phenotypes are environment-dependent and that they change rapidly upon removal from the cellular environment. To overcome this problem, we and others have used in vivo diphtheria toxin-based selective cell ablation [23,24]. Mice expressing Cre recombinase driven by the Kupffer cell-specific Clec4f promoter were crossed with Rosa26-DTR expressing mice to generate KC-specific DTR expression [25]. Sustained KC ablation was achieved by administering DT every 3 days for the final month of the 16-week alcohol exposure, and these mice were compared with similar DT-treated Clec4f-DTR mice on a control chow diet. Preliminary studies using this system have provided new insights into the role of KCs during alcohol consumption [26]. In the control, the chow-fed mice, there was no obvious effect of KC ablation. The mice appeared healthy, there was no weight loss, no induction of liver inflammation or fibrosis, and no loss of liver function. In the alcohol-fed mice, sustained KC ablation produced very different results. There was an increase in the IM content of the liver, an increase in inflammatory cytokine expression and ALT, an increase in hepatic fibrosis, and a loss of differentiated liver functions manifested by an increased serum bilirubin, increased PT/INR, and a decrease in expression of liver-specific genes such as albumin. These results suggest that Kupffer cells play a broad protective role in ALD. At first glance, this result appears to contradict the studies from more than 20 years ago that showed Kupffer cell ablation ameliorating alcohol-induced liver injury [2,5]. However, those earlier studies used ablation methods that killed both KCs and IMs. To try to clarify this discrepancy, we used a global macrophage-targeting method, LysM-Cre-dependent DTR expression, to ablate the total liver macrophages, both KCs and IMs. When fed alcohol, these global macrophage-depleted mice developed reduced inflammation compared to wild-type alcohol-fed mice. Thus, the effects of selective KC depletion are dramatically different than global macrophage depletion, demonstrating a primarily hepatoprotective effect of KCs in a mouse model of ALD.

In summary, liver macrophages are critical components of the liver’s response to alcohol. Their functions are illustrated in Figure 1. Upon alcohol exposure, major changes occur in hepatic macrophage populations. Embryonic Kupffer cells undergo apoptosis and are partially replaced by monocyte-derived KCs, which accounts for the increased heterogeneity of the KC population. At the same time, the quantity of infiltrating macrophages within the liver greatly increases. The combination of in vitro analysis, single-cell gene expression analysis, and selective cell ablation all show that the infiltrating macrophages are the primary inflammatory cells that drive liver inflammation and injury. While there are minor pro-inflammatory KC subsets, the bulk of KCs are hepatoprotective, and they appear to oppose inflammation and fibrosis and preserve hepatocellular function. Selective removal of KCs without removing IMs from the alcohol-exposed liver can result in liver failure, while the removal of both KCs and IMs together reduces liver injury.

## 3. The Role of T Cells in the Immunopathogenesis of ALD

The association between excessive alcohol consumption and increased susceptibility to infection has been documented since the late 1700s [27,28]. This relationship is now firmly established, and severe ALD is well-known to be an immunocompromised state, rendering patients highly vulnerable to overwhelming bacterial infections [29,30,31]. As the first line of defense, the innate arm of the host immunity is the vanguard to bacterial pathogens [32], and this may reflect the extensive research efforts in characterizing the dysfunctional innate immunity in ALD [33,34]. Notably, increasing evidence reveals the key role of adaptive immunity in the antimicrobial armamentarium [35], and particularly, T cells are thought to be central [36]. Multiple studies reveal a consistent failure of the T-cell response in patients with alcohol-related liver disease [37]. Both the quality and quantity of the anti-bacterial T-cell responses are diminished in ALD, characterized by lymphopenia [38], increased levels of T-cell apoptosis [39], an altered balance between T-cell subtypes [40,41], and a reduction in the migratory capability of T cells [42]. As previously shown, the T-cell cytokine production is skewed in ALD, with a loss in the frequency of T-cells producing antibacterial IFNγ in response to the bacterial challenge and the dominance of immunosuppressive IL10 [43]. The balance between IL10 and IFNγ is crucial to allow an appropriate state of host immunity and pathogen defense. IL10 directly impedes pathogen clearance through the potent inhibition of T-cell, monocyte, and neutrophil functions, while conversely, IFNγ is a potent activator of these activities. When this equilibrium is skewed toward IL10, it enables the establishment of an immunological landscape that promotes the persistence of infection [44]. Multiple mechanisms have been suggested to underpin the malfunction T-cell response in ALD, including ethanol-mediated alterations in T-cell metabolism [45,46] and increased T-cell immunosenescence [47]. Our data reveal the role of checkpoint receptors in mediating the impaired anti-bacterial T-cell responses in ALD. These checkpoints are regulatory receptors, found on the surface of immune cells as soluble forms, which keep the immune response ‘in check’. Acting as gatekeepers, they ensure that the immune response, when prompted by infection, is effective but not excessive. They maintain the homeostatic equilibrium between protective antimicrobial immunity and immunopathology [48]. The expression of inhibitory checkpoint receptors to constrain T-cells responses is, in turn, defined by the environment. During inflammation and in the presence of high levels of antigenemia, inhibitory checkpoint receptors are upregulated to ‘switch off’ immune responses, so to limit excessive immunopathology. However, a persistent hyper-expression of the inhibitory checkpoint receptors leads to immune ‘‘exhaustion”, which is associated with a sequential loss of immune activities, including T-cell proliferation, secretion of cytokines, cytotoxic functions, and priming of the pro-apoptotic pathways, causing a progressive immune shutdown [49,50,51,52]. In 2015, we showed the involvement of two inhibitory checkpoint receptors in impairing anti-bacterial T-cell responses in alcoholic hepatitis (AH), namely, programmed cell death 1 (PD1) and T-cell immunoglobulin and mucin domain-containing protein 3 (TIM3). Importantly, we demonstrated that the observed anti-bacterial dysfunction in AH was not permanent but reversible through an ex vivo blockade of PD1/TIM3, and we showed that favorable IFNγ antibacterial responses could be restored and that neutrophil antimicrobial functions could be augmented [43]. Indeed, checkpoint receptor blockade is proving to be effective at rescuing deranged/exhausted immunity in cancer, including hepatocellular carcinoma, and has obtained FDA approval for restoring anti-tumor immunity, with improved response rates and good safety profiles [53,54]. Following on from studies revealing the role of checkpoints in nosocomial infections and septic shock [55], clinical trials of PD1 blockade in bacterial sepsis have found it to be well-tolerated and associated with immune restoration [56,57]. Their clinical safety and efficacy in the context of ALD is yet to be explored. To add a further layer of complexity, we have also described the involvement of the soluble forms of checkpoint receptors in promoting immune paresis in ALD, and these may also confer therapeutic utility [58].

Recent evidence has described the role of unconventional T-cell populations in the deficient antimicrobial response in ALD, including the mucosa-associated invariant T (MAIT) cells, CD1-restricted T cells, and γδ T cells that utilize MR1, CD1 molecules, and BTN/BTNL molecules to respond to lipid, metabolic, or other antigenic stimuli [33]. We have focused on the MAIT compartment: these innate-like CD161-positive T cells are fundamental to the immune control of gut microbiota, bacterial infection, and inflammatory diseases. Initially characterized in the intestinal mucosa, they are the most prevalent population of intrahepatic T cells and exist in high frequencies in the systemic circulation. In response to riboflavin metabolites of bacterial origin, MAIT cells perform their antibacterial functions by secreting cytokines (IFNγ/TNFα/IL-17) and killing infected cells [59]. We found a broad spectrum of dramatic quantitative and functional impairments of blood MAIT cells in ALD patients, which was driven by the loss of gut integrity and increased translocation of gut bacteria to the systemic circulation. MAIT cells were found to be in a state of ‘poised’ activation, displaying an increased expression of activation markers, but lacking in lineage-specific transcription factors and having significantly compromised antibacterial cytokine and cytotoxic responses [60].

The most common cause of death in patients with ALD is the development of serious bacterial infections, and whilst the current guidelines recommend intensive and early antibiotic therapy, this has led to the development of multidrug-resistant bacteria [61]. These infections are associated with a higher incidence of septic shock and/or rapid deterioration of the liver function and death. As such, there is a pressing need to explore new paradigms for anti-infective therapy, and host-directed immunomodulatory therapies are a promising approach. The extensive array of defective antimicrobial T-cell responses in ALD significantly contributes to the immunodeficiency observed in these patients and offers new opportunities for therapeutic intervention [62]. This paradigm is illustrated in Figure 2.

### 3.1. ALD and HSC Activation

HSCs play a leading role in liver fibrosis development and are highly instrumental in the progression of ALD to end-stage liver disease [63]. Following prolonged liver injury, HSCs are activated to become myofibroblasts producing an extracellular matrix (ECM). ECM is indispensable for liver regeneration, but its overproduction and inability to destroy an excessive ECM promotes fibrosis. HSC-activating factors include the induction of inflammatory cytokine cascades, oxidative stress, metabolic reprogramming via the upregulation of autophagy and endoplasmic reticulum stress, and iron overload [64,65,66,67]. These cytokines are secreted by immune cells such as innate lymphoid cells, KC, and bone-derived macrophages [64]. The interaction between HSCs and immune cells regulates the progression of alcohol-associated liver fibrosis via the suppression of NK and T cells by alcohol [68]. Oxidative stress induced by acute and chronic ethanol administration increases TGFβ production in HSCs, followed by an activation of collagen genes, with the further perpetuation of activated HSC phenotypes through ECM remodeling [69].

HSCs contribute to alcoholic steatohepatitis by releasing chemokines and proinflammatory cytokines, such as MCP1, TNFα, and IL-6, as well as latent TGFβ, which suppresses STAT1-activated apoptosis in HSCs [70,71]. In addition, HSCs induce alcohol-associated liver steatosis via triggering the hepatocyte cannabinoid receptor CB1R by HSC-derived endocannabinoid, therefore regulating SREBP1-dependent lipogenesis [72,73]. Importantly, not only can damaged hepatocytes activate HSCs, but HSCs can also program hepatocytes for de novo lipogenesis in response to alcohol-induced hepatic cysteine deficiency and glutathione depletion, as a result of the methionine cycle disruption [74]. The crosstalk between hepatocytes and HSCs is established not only via contact cell-to-cell interactions or cytokine/chemokine production, but via the release of extracellular vesicles (EV), such as exosomes and apoptotic bodies, which are known to induce liver fibrosis [75]. In fact, hepatocyte apoptosis and the release of apoptotic bodies have been demonstrated as an important factor that induce HSC pro-fibrogenic activation [76].

The role of HSCs in liver fibrosis development, with an emphasis on alcohol-related molecular interactions, retinol metabolism, and signaling pathways has already been reported in many excellent review articles [63,72,77,78,79]. However, liver fibrosis pathogenesis may be exacerbated by simultaneous exposure to many “second hits”. Here, we choose to elaborate on less-known aspects of alcohol-induced liver fibrosis, namely, on the potentiation of the effects of alcohol metabolism by hepatocyte infection. In this regard, some mechanisms of HIV-induced liver fibrosis development in the context of alcohol exposure will be overviewed as a clinically relevant but under investigated problem, leading to end-stage liver disease progression triggered by alcohol abuse in the HIV-infected patient cohort.

In addition to infections, the last part of this review will also disclose the role of another second hit, gender, which also regulates the behavior of major NPCs in ALD.

### 3.2. Alcohol Metabolites Affect the Crosstalk between Hepatocyte and Hepatic Stellate Cells to Facilitate Liver Fibrosis Progression: Potentiation by Infectious Agents (HIV)

Alcohol-induced liver damage is facilitated by many infections. While a lot of publications indicate the role of already characterized viral hepatotropic infections (such as HCV and HBV) exacerbated by alcohol abuse, almost nothing is known about Human Immunodeficiency Virus (HIV), which in combination with alcohol, induces significant liver injury with the progression to liver fibrosis [80].

HIV remains a global threat, with approximately 38.4 million active infections and 40.1 million HIV-related deaths [81]. While many may be tempted to think of HIV as a relic of the past, the emerging data suggests otherwise. By the end of 2021, approximately 1.5 million HIV incidences and 650,000 mortalities were reported [82]. There are many reasons for HIV-related mortality, and liver failure is one of them. Liver disease is among the leading organ injuries related to HIV-induced mortality, especially, with the link to antiretroviral therapy-induced longevity among people living with HIV (PLWH) [83,84]. While co-infections of HIV with hepatotropic viruses notoriously contribute to the frequently observed liver disease in HIV-infected individuals [85], alcohol abuse is another significant trigger of liver disease [81]. This is because hepatocytes are the primary site for ethanol metabolism. In addition, alcohol abuse is twice more frequent among people living with HIV (PLWH) than among HIV-uninfected individuals [80]. Moreover, the pathomechanisms of alcohol-induced liver damage among PLWH are not quite clear, while the outcome of the disease to liver fibrosis is quite frequent. Hence, this section of the review aims to highlight the role of hepatic HSCs (as major inducers of liver fibrosis) in alcohol- and HIV-related liver injury.

It is known that interactions between injured hepatocytes and HSCs can promote fibrosis development [86,87]. There are various ways by which damaged hepatocytes can induce HSC profibrotic changes, such as via cytokine/protein release [88], acetaldehyde-malondialdehyde hybrid adducts [89], exosomes [90], and apoptotic bodies [91]. In fact, alcohol exposure can induce hepatocyte apoptosis, which plays a profibrotic role in HSCs [92]. However, the intensity of apoptosis is moderate in alcoholic hepatitis, which might be enhanced by the combination of several triggers of cell death. Hepatotropic viruses are the potent triggers of ethanol-induced liver injury [93]. In fact, it has been shown that the combination of HCV with ethanol induces potent apoptosis, and the engulfment of the formed apoptotic bodies (ABs) by HSCs mediates fibrosis development [94]. This is confirmed by epidemiological data indicating a higher frequency of liver fibrosis in HCV+ alcohol-exposed patients than in alcoholic hepatitis patients [95]. While HCV that is potentiated by alcohol is an already characterized mechanism of liver fibrosis development, studies are very limited in HIV+ alcohol, where the frequency of fibrosis development is also high. This made us initiate this section with an overview of how HIV combined with alcohol to trigger hepatocyte apoptosis, followed by HSC profibrotic activation after the internalization of hepatocyte ABs.

### 3.3. Hepatic Apoptotic Bodies as Activators of Liver Fibrosis under HIV-Alcohol Exposure

According to liver homeostasis, hepatocyte apoptosis usually correlates with HSC profibrotic activation. This phenomenon is profoundly expected, given the high regenerative ability of the liver. However, HSC profibrotic activation may not lead to any major liver impairment if only a few hepatocytes undergo apoptosis. This may be different for massive acetaldehyde production in HIV-infected hepatocytes. In fact, we previously reported on massive hepatocyte apoptosis induced by the combined treatment of hepatocytes with ethanol metabolites (acetaldehyde) and HIV [96]. Hence, hepatocyte apoptosis in the presence of ethanol serves as an acetaldehyde-mediated HIV clearance. However, profibrotic genes were activated in HSCs when HIV-bearing ABs were internalized by HSCs [91]. Therefore, it becomes expedient to understand the mechanisms of acetaldehyde and HIV-induced hepatocyte apoptosis.

HIV is acceptably hepatotoxic [96], but not generally considered hepatotropic. However, several studies have detected HIV in the liver [97,98,99]. The affinity of activated HIV-infected T lymphocytes to the liver may provide some explanation [100]. Moreover, the liver’s anatomical proximity to the gut [101], the largest HIV reservoir, is another important rationale for HIVs presence in the liver [102]. While the aforementioned reasons may support the presence of HIV in the liver, HIV entry into CD4-negative hepatocytes is required for HIV-induced pathogenicity in the liver. Unlike CD4-positive immune cells, HIV endosomal internalization into non-permissive CCR5/CXCR4-rich hepatocytes mediates HIV entry [103]. While this only supports low-level HIV entry [104], a significant amount of HIV accumulates in hepatocytes in the presence of a second hit, such as ethanol and its metabolite, acetaldehyde [96]. The mechanisms of acetaldehyde-induced HIV accumulation, which resulted in hepatocyte apoptosis, can be explained by two significant factors: (1) acetaldehyde-induced alkalinization of the hepatocyte endosome [105,106] and (2) the generation of reactive oxygen species [85].

### 3.4. Alcohol-Induced Endosomal Alkalinization Supports HIV Accumulation in Hepatocytes

Hepatocytes are considered non-permissive to HIV because they lack CD4. However, hepatocyte richly expresses HIV-co-receptors, CCR5, and CXCR4 [96,107], which interact with HIV envelope glycoproteins for HIV entry. Moreover, HIV entry through endocytic internalization has also been described as an alternative HIV entry pathway for CD4-negative cells [103]. While the endocytic HIV entry pathway is a non-canonical HIV entry mechanism, substantial supporting evidence for this pathway is available. Marechal et al. demonstrated receptor-mediated endocytosis of HIV in monocyte-derived macrophages [108]. Similarly, another study suggested endocytosis as the mechanism for HIV entry into T cells and monocytes after blocking CD4 receptors with antibodies [109]. A recent study also validated endocytosis as the HIV entry mechanism for CD4-negative cells [103]. Hepatocytes that are CD4 negative are therefore, not exempted from the endocytic internalization of HIV. In fact, we demonstrated the HIV endocytic internalization in hepatocytes [90].

Meanwhile, antigens that are internalized through the endocytic pathways are canonically fated for degradation by the pH-dependent (acidic) endosomal/lysosomal system [110]. This may explain why only low-level HIV is detected in intact hepatocytes [96,104], but due to acetaldehyde-dependent alkalinization of endosomal compartments, more HIV survives in alcohol-exposed hepatocytes. However, HIV survival in the endosome may be sustained when the lysosome degradation function is impaired. In fact, bafilomycin-mediated lysosome alkalinization, which impaired lysosome functions, increased HIV infectivity in HeLa Magi cells [111]. Similarly, alcohol is known to alkalinize lysosomes [106], resulting in prolonged survival of HIV in the endosomal/lysosomal system, where they were fated for degradation [90]. This explains why we observed an accumulation of HIV gag RNA and proteins in hepatocytes exposed to acetaldehyde [96].

### 3.5. Alcohol-Induced HIV Accumulation Triggers Reactive Oxygen Species Generation

Substantial evidence exists to support HIV as a potent trigger for ROS generation, both in vivo and in vitro. Elbim et al. observed a positive correlation between HIVs viral load and ROS generation [112]. Another salient observation was the ROS generation due to acetaldehyde-induced HIV accumulation in hepatocytes [96]. A clinical study observed a reduction of thioredoxin and an abundant antioxidant in the lymphoid tissues of AIDS patients [113]. The elevation of lipid peroxidation products in the serum of HIV-infected individuals, compared to the controls, is another valid clinical evidence of HIV-induced ROS generation [114]. Staal et al. also observed glutathione and cysteine depletion among AIDS patients [115].

In addition, HIV proteins have been reported as a modulator of HIV-induced ROS generation. For example, HIV TAT (trans-activator of transcription) was observed to induce hydrogen peroxide [116]. HIV gag protein, p24, is another HIV protein that triggers ROS generation, as detected by 2′,7′-dichlorodihydrofluorescein [96]. Other studies revealed gp120, an HIV envelope protein, to be a trigger of oxidative stress [117,118]. Viral protein R (Vpr) is also an HIV protein that has been shown to induce oxidative stress in Schizosaccharomyces pombe cells [119]. While most HIV proteins trigger ROS generation, they do this through various mechanisms. For example, ROS generated by HIV TAT in ECV-304 cells was attenuated by NADPH oxidase inhibitors. Hence, NADPH is the mechanistic pathway for HIV TAT-induced ROS generation [120]. Mitochondria is another organelle involved in the HIV-induced ROS generation, particularly by Vpr [121] and HIV TAT [122]. The activation of NOX4 by the Vav/Rac/PAK pathway has also been observed by HIV Nef [123]. In addition to HIV-mediated ROS release, ROS may be released by host cells as a defensive mechanism against HIV [124].

Whether ROS is mediated by cellular defense mechanisms or as a direct effect of HIV pathogenicity, it leads to oxidative stress and cell death. In fact, immune cell depletion, which is a typical feature of HIV pathogenesis, is triggered by ROS [125]. Similarly, we observed HIV-induced death in hepatocytes exposed to acetaldehyde in CYP2E1-overexpressing liver cells [126]. While ethanol and acetaldehyde trigger the generation of ROS in HIV-containing hepatocytes, they also induce HIV accumulation via a change in the lysosomal pH, leading to oxidatively induced apoptosis [90]. Recent studies showed that *N*-acetylcysteine, a potent antioxidant, can reverse hepatocyte oxidative death by the restoration of lysosome functions [126]. This suggests the involvement of ROS-induced lysosome impairment in hepatocyte apoptosis. In fact, our data revealed crosstalk between lysosome and mitochondria as the mechanisms for ROS-induced hepatocyte apoptosis [126]. While this process seemed beneficial since it provided clearance of HIV-accumulated hepatocytes, it is also a detrimental event. This is because the hepatocyte-derived ABs, when engulfed by HSCs, activate profibrotic genes in these cells, while the activation of pro-fibrotic genes was not observed when ABs generated from HIV-infected lymphocytes were internalized by HSCs [96].

### 3.6. Apoptotic Bodies Derived from HIV and Acetaldehyde-Exposed Hepatocytes Induce HSC Profibrotic Activation

HSC activation after ABs engulfment has previously been observed [127]. Moreover, as demonstrated by comparing the pro-fibrotic effects of ABs derived from hepatocytes vs. ABs derived from lymphocytes, only the ABs of hepatocyte origin could activate HSCs [96]. Beyond the cell origin of ABs, the mechanisms of HSCs activation may depend on the type of agent that induces apoptosis. For example, studies that utilized UV as an apoptotic trigger indicate Toll-like receptor (TLR)-9 as the mechanistic pathway for HSC activation [128]. This may be attributed to the ability of UV to efficiently disintegrate DNA into the CpG motifs required to activate the TLR9 [129]. However, this is not the case when both acetaldehyde and HIV induce apoptosis of hepatocytes, as we recently demonstrated in vivo [91]. Therefore, it becomes paramount to decipher the contents of hepatocyte ABs generated from the combined treatment with acetaldehyde and HIV (AB_AGS+HIV_). An enormous amount of HIV proteins and an oxidative product, malondialdehyde (MDA), were observed as part of AB_AGS+HIV_ cargo [91], and these AB_AGS+HIV_ express phosphatidylserine, which acts as the “eat me” signal for HSCs [130].

To complement the aforementioned AB_AGS+HIV_ characteristics, HSCs were found to express ligand bridge proteins, Gas6, ProS, AXL, and phosphatidylserine recognition receptor [91]. Interactions between the ligand bridge proteins and AXL mediates AB_AGS+HIV_ entry into HSCs [91]. Since internalized AB_AGS+HIV_ contains HIV proteins and MDA, the HSC activation should highlight the pathways that are mediated by oxidative stress and/or HIV proteins. In our recent study, JNK inhibitors attenuated HSC profibrotic activation by ABs via the ERK1/2 pathway [91]. Likewise, *N*-acetyl cysteine attenuated ERK1/2 and HSC profibrotic activation [91]. This suggests that profibrotic activation in HSCs is partly triggered by oxidative stress through the JNK-ERK1/2 pathway. Given that an oxidative product, 4-Hydroxy-2, 3-nonenal, activated the JNK pathway in the study of Parola et al. [131], MDA from hepatic AB_AGS+HIV_ may have triggered the observed JNK-ERK1/2 pathway. In addition, the JAK-STAT3 pathway is another ROS-dependent pathway for HSC profibrotic activation [132]. Our data demonstrated HSC profibrotic attenuation by an ROS-triggering hepatic ABs in a STAT3-silenced HSC. This confirms the involvement of the JAK-STAT3 pathway after AB_AGS+HIV_ engulfment by HSCs, while the JAK-STAT1 pathway was suppressed [91].

While hepatocyte apoptosis provides the premises for HIV and acetaldehyde/ROS-induced hepatic fibrosis, inhibiting hepatocyte apoptosis may seem like the best target for clinical intervention. However, this may have detrimental consequences, since acetaldehyde-induced hepatocyte apoptosis is an avenue for HIV clearance from these liver cells, and the suppression of this clearance leads to increased HIV DNA expression and even HIV DNA integration into the human genome, as has been shown by treatment cells with pan-caspase inhibitor [96]; the profibrotic effects of the MDA and HIV-containing AB_AGS+HIV_ can be attenuated by augmenting antiretroviral therapy by liver-targeted antioxidants.

### 3.7. Sex Differences in NPC Properties in ALD

Here, we present the data on gender-dependent regulations of NPC functions. Along with infection, gender is considered as a second hit for ALD that regulates its progression. In fact, it has long been recognized that the consequences of alcohol consumption are different in males and females [133,134,135,136,137]. Men have lower median platelet counts and higher serum creatinine, ALT, and GGT concentrations [133]. A small study indicated that females have higher mortality in acute alcoholic hepatitis [133]; however, later data from TREAT consortium did not find significant differences between genders [138]. These discrepancies could be explained by the fact that the diagnosis of AH in most cases was made based on a combination of clinical and laboratory data and without histological confirmation. Several studies indicate that alcohol metabolism is different between males and females [136,137,139]. Other studies have noted differences in hepatocyte proliferation/regeneration and alcohol-induced hepatocyte apoptosis [140]. Gonadectomy experiments suggest that these pathways are in part, regulated by sex hormone signaling [141]. Some of the differences were reported to be dependent on the sex-specific hormone milieu of the animal, but others persist even in cells isolated from males and females, independent of endogenous hormones [136]. Thus, sex differences in non-parenchymal cell signaling in ALD are influenced by altered hepatic metabolism of alcohol as well as sex differences in non-parenchymal cells themselves. Recent studies revealed that males’ and females’ liver disease progression differ in innate and adaptive immunity, fibrosis signaling, and other pathways that involve multiple NPC populations [135,141,142,143,144]. Single-cell RNA-sequence studies confirmed that the alcohol effect on every liver cell population was indeed sex-specific [145].

### 3.8. Sex Differences in Macrophage Properties

Female livers have a greater number of liver macrophages (Kupffer cells) per gram tissue [146]. In addition, a lower percentage of Kupffer cells is present in the vicinity of hepatic stellate cells (HSCs) in female livers compared with males [147], and therefore, less fibrotic tissue is present under normal conditions.

After alcohol exposure, females exhibit injury more quickly than males. Moreover, levels of nuclear factor kappa B are doubled in female livers compared with male livers after ethanol treatment [148]. The expression of MyD88, a downstream signaling molecule or TLR signaling, was only found to be significantly induced in the livers of female alcohol-exposed mice [149]. These differences suggest higher sensitivity of female liver macrophages to endotoxin/LPS. In fact, in vivo estrogen treatment increases the sensitivity of hepatic macrophages to endotoxin [148]. Another study confirmed that estrogen has a major influence on the susceptibility of Kupffer cells to gut-derived LPS, resulting in increased proinflammatory cytokine production, which could be a major contributing factor to the increased risk of alcohol-associated liver disease in women [150]. Anti-estrogen (toremifene) treatment reduced the effects of alcohol in females but had no effect on the production of TNF-alpha by isolated Kupffer cells or liver inflammation.

Several studies reported sex differences in chemokine induction by alcohol treatment. Higher CCL-2 chemokine levels produced by liver macrophages in response to alcohol in females could explain greater injury in females, since the *Ccl2* gene deficiency protects mice against alcohol-associated liver injury [151]. Other studies report that sex differences in chemokine induction by alcohol treatment were dependent on diabetic condition [152], suggesting that there is a complex interplay between sex and alcohol response.

Some studies have noted that, in addition to higher sensitivity to endotoxin, females have higher levels of endotoxemia [148,153].

Hepatocyte-produced oxysterols are crucial for maintaining Kupffer cell identity and phenotypes through LXRα signaling [154]. Studies of 27-hydroxycholesterol (27-HOC) have demonstrated that this signaling can be sex-specific. Researchers have shown that 27-HOC administration oppositely affected inflammation in female and male macrophages. These sex-opposed inflammatory effects of 27-HOC were shown to be estrogen-dependent [155].

Sex differences in macrophages were reported not only in the liver but also in the adipose tissue [156]. Female mice after alcohol exposure had greater adipose tissue inflammation in vivo, and they showed an increased expression of TNFα and CCL-2, while IL-6 induction in adipose tissue was not sex-specific. These data correlated with an increase in macrophage activation markers and induced expression of TLR receptors in adipose tissue macrophages.

Taken together, macrophages in females demonstrate higher pro-inflammatory signaling in the presence of alcohol, which is likely mediated by estrogen, hepatocyte-derived factors, and alterations in TLR signaling.

### 3.9. Sex Differences in T-Cells Functions

It is well-known that many T-cell subpopulations are altered in patients with alcohol use disorders (AUD) and ALD. Several of these changes are sex specific. Almost one out of four patients with AUD have high CD4 and CD8 double-positive T cells, and the frequency of this event was three times more in women than in men [157]. CD4+ CD8+ double-positive T cells have been associated with autoimmune diseases and inflammation, suggesting that alcohol may modulate liver T-cell responses in a sex-specific way. A study of the T-cell population in the livers of ALD patients confirms that higher Th17 and lower T-regs in ALD patients correlated with poor survival. This study noted that males with ALD had significantly lower T-regs, while females showed a higher pro-inflammatory cytokine production that was correlated with complications and a poor 90-day outcome [158].

Another study suggested that T-cell differences could be attributed to differentially regulated signaling pathways in the dendritic cells in females compared to males. These differences were exacerbated by ethanol treatment [159]. Female dendritic cells treated with ethanol were unable to activate antigen-specific cytotoxic T cells (CTLs), as shown by the reduced expression of CD44, CD69, and the decreased production of IFNγ.

While recent studies have uncovered the role of Th17 signaling in ALD progression [160] and previous studies have demonstrated the sex-specific role of IL-17A in various diseases [161,162,163], more investigations of sex-specific pathways in T-cell responses in the liver are necessary.

### 3.10. Sex Differences in HSC Properties

Very few studies explored sex differences in HSCs in alcohol-induced liver disease. However, some results from other disease models indicate that HSC signaling is sex de-pendent. NASH models suggest that the estrogen-mediated crosstalk between hepatocytes and HSCs may contribute to sex differences in non-alcoholic fatty liver disease through an anti-fibrogenic function of the sphingosine 1-phosphate [164]. Several studies have reported that estrogen therapy improved hepatic fibrosis and inhibited the activation of HSCs [165,166]. While some studies reported a direct effect of the estrogen receptor signaling, others indicated that active estrogen metabolites, with little or no affinity for ERα and ERβ, could mediate the anti-fibrotic effect of estrogens through ER-independent pathways [166]. In vitro studies reported that in cultured HSCs, estradiol-inhibited type I collagen production, alpha-SMA expression, and cell proliferation [167]. These findings seem to contradict the notion that females are more susceptible to alcohol-induced fibrosis progression and at a higher risk of developing cirrhosis, independent of disease severity [133,168,169]. The data suggest that more complex molecular mechanisms are involved in sex-differences in HSC activation that is induced by alcohol.

A recent study identified histone demethylation enzymes, KDM5B and KDM5C, which have sex-specific roles in HSCs after alcohol exposure. In female mice, KDM5B and KDM5C promoted alcohol-induced HSC activation and fibrosis development in mice and humans, while in males, HSC activation was KDM5B and KDM5C independent [170]. This study also suggested that sex differences were mediated by an estrogen-dependent mechanism that involved female-specific *Ahr* and *Arnt* transcriptional repression by KDM5B and KDM5C, and AhR pathway inhibition in activated HSCs [170]; however, other mechanisms were not excluded.

Taken together, there are pro- and anti-fibrotic mechanisms involved in HSC sex differences in ALD.

## 4. Conclusions

In ALD:There is a replacement of embryonic KC with monocyte-derived KC. KCs mainly play a hepatoprotective role and participate in phagocytosis and matrix remodeling, while a minor part is pro-inflammatory. Inflammation is related to infiltrating macrophages that cause hepatocyte dedifferentiation and fibrogenesis. In ALD, the selective removal of KCs without removing IMs from the alcohol-exposed liver can result in liver failure, while the removal of both KCs and IMs together reduces liver injury. This is illustrated by Figure 1.Impaired anti-bacterial protection is due to dysfunctions in the innate immunity and T cells based on depleted T-cell frequencies, increased cell apoptosis, imbalanced T-cell subsets, impaired production of cytokines, and the reduced ability to kill bacteria (Figure 2).The significant trigger for profibrotic activation is the engulfment of apoptotic bodies by HSCs, which is induced by ethanol metabolites and can be further potentiated by viral infections, including HIV. This apoptotic body formation is regulated by the induction of oxidative stress due to the lysosome dysfunction-dependent accumulation of HIV proteins in ethanol-exposed hepatocytes.There is a sex difference in the regulation of liver NPCs number/functions in macrophages and HSC, while the sex-dependent regulation of endothelial and immune cells requires further clarification.

## Figures and Tables

**Figure 1 biology-12-00255-f001:**
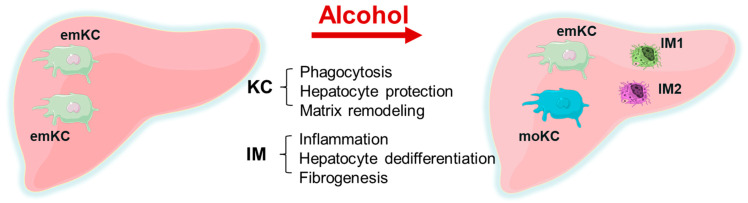
Liver macrophage population changes induced by alcohol. In response to alcohol, there is an influx of monocyte-derived infiltrating macrophages (IM) as well as loss of embryonic Kupffer cells (emKC), with their replacement by monocyte-derived Kupffer cells (moKC).

**Figure 2 biology-12-00255-f002:**
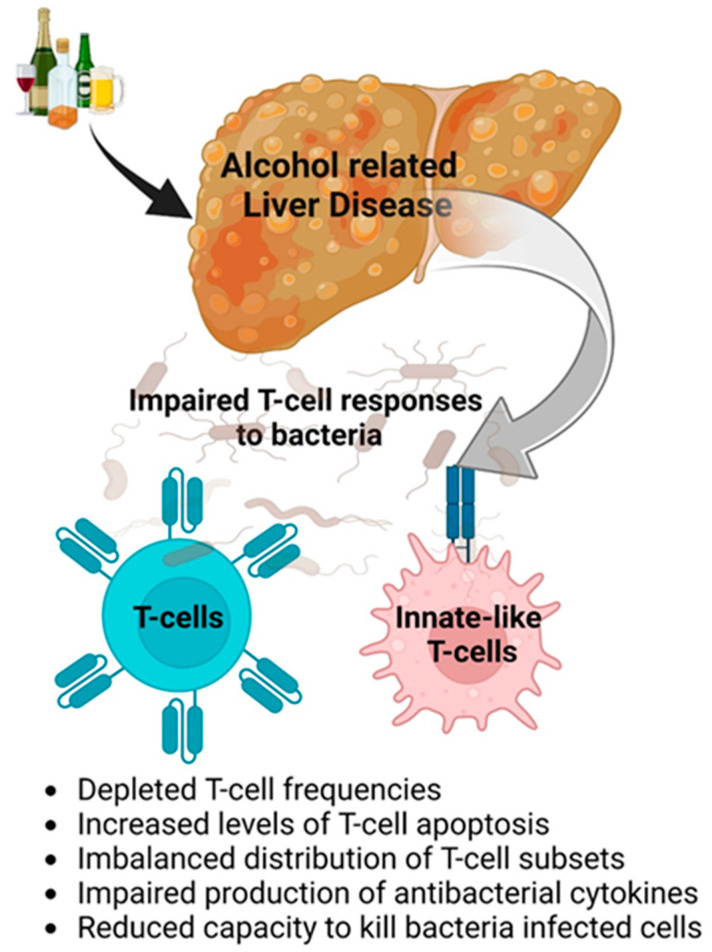
ALD is associated with a dysfunctional T-cell response.

## Data Availability

Not applicable.

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
