# Peer review of "A Pathogenic Role of Non-Parenchymal Liver Cells in Alcohol-Associated Liver Disease of Infectious and Non-Infectious Origin"

_biology, 2023, doi:10.3390/biology12020255_

Round 1
Reviewer 1 Report
The review manuscript by Kharbanda et al. provides an overview of the recent papers focused on the role of NPC in ALD with and without coexisting infections (e.g., HCV and HIV). Review focuses on liver macrophages and T cells in ALD, however hepatic stellate cell role is mostly discussed in HIV-ALD context. There are very important sections in the beginning of the review on sex differences in the properties of macrophages, HSCs, endothelial and T cells in ALD. Remaining review focuses on macrophages, T cells, and HSCs, however HSCs sections are only in the context of HIV. There is also a section on macrophages in other liver disease NAFLD, which is not consistent with the review. Here are other comments:
Sex difference sections could be moved at the end
Authors should add a section on HSCs in ALD without HIV to complete the review of role of NPC in ALD and maybe remove sex differences in endothelial cell function since there is no section for endothelial cells
Lines 129-130: please elaborate what are the roles of these histone demethylation enzymes in HSCs function/activation
Line 242: please clarify what is “this model”
Line 267: lipid associated macrophages should be abbreviated as (LIMs)
Line 482: reference 161 does not report any fibrosis or HSCs
Line 484: Ref 162 is from 1996 and does not mention fibrosis
Line 475: Ref 155 describes the role of HSC-derived mediators, not hepatocyte-derived
Lines 582-583: “Moreover, only Abs of hepatocyte origin could activate HSC”. Even if it is true, authors should provide correct references for this statement
Lines 354-364 are the same as lines 620-629
Lines 438-446 are the same as lines 631-639
Lines 640-646 are the same as 611-619
Reviewer 2 Report
The paper by Kharbanda et al is an interesting review on the participation of non-parenchymal liver cells (NPCs) in alcoholic liver disease (ALD).
The paper, as the title says, deals with the pathogenetic role of NPCs in alcohol related diseases. It is an interesting and ambitious effort, but it has some flaws.
Major Points
The paper requires extensive restructuring. The information provided should be presented in a more reasonable way. Thus, section 2 can be merged with the introduction after proper modifications.
Sections 3,4,5,6 would be better incorporated in the relevant sections. In addition, lines 473-484 should be removed from the HIV section and included in the role of HSC in ALD.
Sections 7 and 10 should be merged as they are referring to the same subject. Moreover, it would be better if sections 8 and 9 were omitted as they are irrelevant to the scope of this review and by themselves could well be the subject of a different extensive review.
The section of HIV as a confounding factor in ALD should be clearer and form a new section (12?) with subsections. In that respect lines 620-639 should be removed and added to the relevant sections of Kupffer cells and T cells respectively.
Minor points
1.Language requires polishing as there are problems in some expressions and spelling. For example, in the first paragraph of the abstract the verb is missing. In line 43: the word “strongest” is not the proper one there. In addition, in line 481 the expression “conspiracy of HCV with alcohol” may be poetic but it is not scientific.
2.Repetition of statements in lines 48-54. The meaning of the two paragraphs is the same.
3.Sex differences mortality. Please comment on the fact that ref 2 refers to a small number of alcoholic hepatitis patients without histological confirmation of the diagnosis.
Round 2
Reviewer 2 Report
The authors have responded to most of the suggestions. The study is now comprehensively presented.